# Nonlinear Simulation and Performance Characterisation of an Adaptive Model Predictive Control Method for Booster Separation and Re-Entry

**Joseph Chai** [1] **and Erkan Kayacan** [2,*]

1   School of Mechanical and Mining Engineering, University of Queensland, Brisbane 4072, Australia
2   School of Aerospace and Mechanical Engineering, University of Oklahoma, Norman, OK 73019, USA
*   Correspondence: erkan@ou.edu

**Abstract:** This paper evaluates the $\mathcal{L}_1$ adaptive model predictive control (AMPC-$\mathcal{L}_1$) method in terms of its control performance and computational load. The control performance is assessed on the basis of the nonlinear simulation of a fly-back booster conducting stage separation and re-entry, and compared to baseline nonadaptive MPC and as a pole placement controller in both longitudinal and lateral control tasks. Simulation results show that AMPC-$\mathcal{L}_1$ exhibits superior control performance under nominal conditions, and aerodynamic and guidance law uncertainties. The computational load of AMPC-$\mathcal{L}_1$ is also evaluated on an embedded platform to demonstrate that AMPC-$\mathcal{L}_1$ preserves the efficiency properties of AMPC while improving its performance.

**Keywords:** adaptive control; model predictive control; $\mathcal{L}_1$ adaptive control; re-entry vehicle

## 1. Introduction

Adaptive control and model predictive control (MPC) are two historically separate areas of control theory that provide different performance benefits to the control problem of an uncertain system. At each control update, MPC solves a finite-horizon optimal control problem with a receding horizon principle. The solution to the optimal control problem is the control input calculated on the basis of the prediction of future states [1,2]. MPC is a control method with predictive capability and superior constraint handling compared to those of classical PID controllers, thereby improving control performance [3,4]. As a form of optimal control, MPC seeks minimal tracking errors while abiding by the constraints of the system model [5,6]. Moreover, the predictive element of MPC means that the controller acts pre-emptively to squash tracking errors instead of waiting to react to errors as in PID control. An excellent review of MPC studies in aerospace systems was provided by Eren et al. [7]. With respect to re-entry vehicles, Van Soest et al. implemented MPC for the re-entry simulation of the X-38 unpowered crew return vehicle (CRV) at 10 Hz [8], and Pascucci et al. implemented MPC for a re-entry vehicle undergoing powered descent using thrust vectoring [9].

The field of adaptive control addresses a different issue. The inherent presence of uncertainties in a system model results in a mismatch between the model that is used to synthesise the control algorithm and the real system, resulting in performance degradation for classical and optimal control methods [10–13]. Adaptive control seeks to compensate for uncertainties in the system model by either (i) estimating the system model explicitly and modifying the control law accordingly or (ii) estimating the representative matched and unmatched uncertainties present in the system, and compensating for them in the control law [14–18]. Both categories aim to retain the nominal design performance of the nonadaptive controller in the presence of model uncertainties.

Adaptive control has seen much activity within the aerospace community. Banerjee et al. applied an $\mathcal{L}_1$ augmentation to a pole placement controller for the longitudinal

dynamics of hypersonic gliders. Simulation studies found that adaptive augmentation improved control performance [19]. An optimal control modification was applied to a model reference adaptive controller (MRAC) for a fighter aircraft [20]. Compared to the baseline controller and simple MRAC, the optimal control modification performed better. Zhou et al. presented a sliding-mode adaptive control algorithm based on a backstepping approach [21]. In that study, the sliding-mode control law was used to suppress the effects of parameter variations and disturbances in the attitude rate loop. Liu et al. designed a high-performance adaptive controller for hypersonic flight vehicles [22]. The simulation results showed that the developed control technique could improve the transient response of hypersonic flight vehicles compared to other adaptive strategies, focusing on parametric uncertainty and asymptotic tracking.

Despite the different historical roots of adaptive control and MPC, there have been many attempts to derive algorithms that combine elements of the two, resulting in adaptive MPC. Various forms of adaptive MPC exist in the literature that have been applied to aerospace control problems. Chowdary et al. propose a concurrent learning algorithm combined with constrained model predictive control [23,24]. Feedback linearisation is used to linearise a nonlinear aircraft wing-rock model. Concurrent learning (CL) is a modified direct model reference adaptive control (MRAC) scheme that relaxes the persistent excitation requirement by utilising the long-term memory of past parameter estimates and tracking errors. The CL controller is used online to identify the plant parameters. Upon convergence, a switching rule is used to switch to an MPC control algorithm that utilises the estimated plant parameters in its optimisation routine. Improvements to roll tracking performance were observed when the concurrent learning converges and MPC is switched on. If the parameters deviate substantially while MPC is switched on, the algorithm switches back to concurrent learning to relearn the correct parameters.

Mehndiratta et al. presented a learning-based adaptive nonlinear MPC (NMPC) implementation for a ground robot and a small unmanned aerial vehicle (UAV) with experimental results [25]. The learning algorithm was a nonlinear moving horizon estimator (NMHE) that estimated the state variables and plant parameters. The estimated state and parameters were then used in the NMPC algorithm to compute the optimal control input. For the UAV experiment, the NMHE estimated the velocities and mass of the vehicle. The estimated velocities and mass were used in the position hold NMPC to compute the desired vertical force and attitude angles. Experimental results showed that, with the NMHE updating the parameter estimates, the UAV tracked the desired trajectory with less tracking error than that in the case with NMHE off, especially in altitude.

Pereida and Schoellig presented an adaptive MPC scheme using $\mathcal{L}_1$ adaptive control for the trajectory tracking of an agile quadrotor [26]. The implementation of the $\mathcal{L}_1$ adaptive control method estimated the matched uncertainties of the system using an adaptive law, but did not account for unmatched uncertainties. MPC was used to generate reference commands that were augmented with the $\mathcal{L}_1$ adaptive control law. This resulted in improved experimental performance in contrast to that of nonadaptive and nonpredictive methods despite dynamic environmental disturbances.

While MPC provides many performance benefits, autonomous aerospace vehicles impose form factor requirements on computing hardware, which has led to a drive for efficient control algorithms. This has led to research in reducing the computational complexity of MPC. Explicit MPC involves precomputing solutions to optimal control problems offline and storing them into multidimensional look-up tables. Explicit MPC was applied to a spacecraft control problem [27]. Another approach to reducing the computational requirements of MPC is to reduce the number of prediction points over the finite horizon. Algebraic MPC is a form of efficient MPC first proposed by Gibbens and Medagoda to reduce the computational burden of conventional MPC by replacing the Taylor series expansion of the state transition matrix with an eigendecomposition, allowing for the exact evaluation of the state transition [28]. This eliminates the truncation error associated with

large time steps in Taylor series approximation, thereby allowing for a reduced number of prediction time steps without the usual penalties.

Chai et al. proposed an adaptive and efficient MPC scheme based on algebraic MPC and $\mathcal{L}_1$ adaptive control, and applied the algorithm on a fly-backk booster re-entry longitudinal linear time-varying (LTV) model [29]. AMPC-$\mathcal{L}_1$ combines the efficiency benefits of AMPC with the performance guarantees of $\mathcal{L}_1$ adaptive control in the presence of uncertainties. AMPC-$\mathcal{L}_1$ retains the nominal design performance of AMPC despite disturbances and model mismatch, while baseline AMPC experiences degraded performance. Although Chai et al. presented the derivation for AMPC-$\mathcal{L}_1$ and demonstrated the performance of the algorithm on an LTV model, this paper extends the existing research by: (i) including stage separation in the mission scenario, (ii) using a high-fidelity nonlinear simulation model with both longitudinal and lateral dynamics, and (iii) describing the implementation of AMPC-$\mathcal{L}_1$ on an embedded Linux computer. Specifically, the contributions of this paper are as follows:

- Presenting an AMPC-$\mathcal{L}_1$ controller design for longitudinal and lateral control of a re-entry vehicle undergoing stage separation and re-entry maneuvers.
- Evaluating AMPC-$\mathcal{L}_1$ against baseline AMPC and pole placement through nonlinear simulation of booster separation and re-entry with modeling and input uncertainties.
- Presenting an overview of a software implementation of AMPC-$\mathcal{L}_1$.
- Demonstrating AMPC-$\mathcal{L}_1$ implemented for a representative embedded computer and characterising the computational expense.

## 2. Fly-Back Booster Dynamic Model

The fly-back booster model is based on a dynamic aircraft model provided by Zipfel with different vehicle-specific aerodynamics, gravimetrics, and mission profiles [30]. The fly-back booster is intended as the first stage for a second-stage scramjet-powered accelerator [31]. More details on the development of the fly-back booster model may be found in [32,33].

The system states of a fly-back booster during stage separation and re-entry are position $(s_N, s_E, s_D)$, velocity $(u, v, w)$, attitude $(\phi, \theta, \psi)$, and body rates $(p, q, r)$. The nonlinear system state dynamics equations are as follows:

$$\begin{bmatrix} \dot{u} \\ \dot{v} \\ \dot{w} \end{bmatrix} = -\begin{bmatrix} 0 & -r & q \\ r & 0 & -p \\ -q & p & 0 \end{bmatrix}\begin{bmatrix} u \\ v \\ w \end{bmatrix} + \frac{1}{m}\begin{bmatrix} f_{a,x} \\ f_{a,y} \\ f_{a,z} \end{bmatrix} + T_l^b\begin{bmatrix} 0 \\ 0 \\ mg \end{bmatrix}, \tag{1}$$

$$\begin{bmatrix} \dot{s}_N \\ \dot{s}_E \\ \dot{s}_D \end{bmatrix} = (T_l^b)^T\begin{bmatrix} u \\ v \\ w \end{bmatrix}, \tag{2}$$

$$\begin{bmatrix} \dot{p} \\ \dot{q} \\ \dot{r} \end{bmatrix} = \begin{bmatrix} I_{11} & I_{12} & I_{13} \\ I_{12} & I_{22} & I_{23} \\ I_{13} & I_{23} & I_{33} \end{bmatrix}^{-1}\left(-\begin{bmatrix} 0 & -r & q \\ r & 0 & -p \\ -q & p & 0 \end{bmatrix}\begin{bmatrix} I_{11} & I_{12} & I_{13} \\ I_{12} & I_{22} & I_{23} \\ I_{13} & I_{23} & I_{33} \end{bmatrix}\begin{bmatrix} p \\ q \\ r \end{bmatrix} + \begin{bmatrix} m_{a,x} \\ m_{a,y} \\ m_{a,z} \end{bmatrix}\right), \tag{3}$$

$$\begin{bmatrix} \dot{\phi} \\ \dot{\theta} \\ \dot{\psi} \end{bmatrix} = \begin{bmatrix} 1 & \sin\phi\tan\theta & \cos\phi\tan\theta \\ 0 & \cos\phi & -\sin\phi \\ 0 & \sin\phi/\cos\theta & \cos\phi/\cos\theta \end{bmatrix}, \tag{4}$$

where $I$ is the inertia matrix, $m$ is the mass, $g$ is the gravitational force, $f_a$ is the aerodynamic forces, $m_a$ is the aerodynamic moments, and $T_l^b$ is the transformation matrix from local-level coordinates to body coordinates. Written in full, $T_l^b$ is expressed as follows:

$$T_l^b = \begin{bmatrix} \cos\psi\cos\theta & \sin\psi\cos\theta & -\sin\theta \\ \cos\psi\sin\theta\sin\phi - \sin\psi\cos\phi & \sin\psi\sin\theta\sin\phi + \cos\psi\cos\phi & \cos\theta\sin\phi \\ \cos\psi\sin\theta\cos\phi + \sin\psi\sin\phi & \sin\psi\sin\theta\cos\phi - \cos\psi\sin\phi & \cos\theta\cos\phi \end{bmatrix}. \tag{5}$$

As the booster was unpowered during stage separation and re-entry, no propulsive forces were accounted for in this model. The aerodynamic forces and moments were defined using the following equations:

$$\begin{bmatrix} f_{a,x} \\ f_{a,y} \\ f_{a,z} \end{bmatrix} = \bar{q}S \begin{bmatrix} C_X \\ C_Y \\ C_Z \end{bmatrix}, \text{and} \tag{6}$$

$$\begin{bmatrix} m_{a,x} \\ m_{a,y} \\ m_{a,z} \end{bmatrix} = \bar{q}Sb \begin{bmatrix} C_l \\ C_m \\ C_n \end{bmatrix} \tag{7}$$

where $C_X$, $C_Y$, and $C_Z$ are the aerodynamic force coefficients, and $C_l$, $C_m$, and $C_n$ are the aerodynamic moment coefficients computed on the basis of a combination of look-up tables and aerodynamic derivatives based on a first-order Taylor series expansion. $\bar{q}$ is dynamic pressure, $S$ is the reference area, and $b$ is the reference length for the lateral coefficients. The aerodynamic derivatives were stored in look-up tables as functions of the Mach number ($M$) and angle of attack ($\alpha$). The equations for the aerodynamic coefficients are as follows:

$$C_X = C_{X_0}(M,\alpha) + C_{X_{\delta_e}}(M,\alpha)|\delta_e| + \frac{c}{2V}C_{X_q}(M,\alpha)q, \tag{8}$$

$$C_Y = C_{Y_0}(M,\alpha) + C_{Y_{\delta_r}}(M,\alpha)\delta_r + C_{Y_\beta}(M,\alpha)\beta + \frac{b}{2V}(C_{Y_r}(M,\alpha)r + C_{Y_p}(M,\alpha)p), \tag{9}$$

$$C_Z = C_{Z_0}(M,\alpha) + C_{Z_{\delta_e}}(M,\alpha)\delta_e + C_{Z_\beta}(M,\alpha)\beta + \frac{c}{2V}C_{z_q}(M,\alpha)q, \tag{10}$$

$$C_l = C_{l_0}(M,\alpha) + C_{l_\beta}(M,\alpha)\beta + C_{l_{\delta_a}}(M,\alpha)\delta_a + C_{l_{\delta_r}}(M,\alpha)\delta_r + \frac{b}{2V}(C_{l_r}(M)r + C_{l_p}(M)p), \tag{11}$$

$$C_m = C_{m_0}(M,\alpha) + C_{m_{\delta_e}}(M,\alpha)\delta_e + \frac{c}{2V}C_{m_q}(M,\alpha)q, \tag{12}$$

$$C_n = C_{n_0}(M,\alpha) + C_{n_\beta}(M,\alpha)\beta + C_{n_{\delta_r}}(M,\alpha)\delta_r + \frac{b}{2V}(C_{n_r}(M,\alpha)r + C_{n_p}(M)p), \tag{13}$$

where $V$ is the vehicle velocity magnitude, $c$ is the reference area for longitudinal coefficients, and $\beta$ is the side-slip angle. The control inputs are aileron ($\delta_a$), elevator ($\delta_e$), and rudder ($\delta_r$).

## 3. Controller Design

In this section, the controllers are presented for both the longitudinal and the lateral cases using the AMPC-$\mathcal{L}_1$ control algorithm, which consists of a state predictor, the $\mathcal{L}_1$ adaptive law, the $\mathcal{L}_1$ control law, and the AMPC optimal control law.

### 3.1. Summary of AMPC-$\mathcal{L}_1$ Algorithm

An overview of the AMPC-$\mathcal{L}_1$ architecture is shown in Figure 1. The details of the AMPC-$\mathcal{L}_1$ algorithm are found in [29]. The dashed box shows the $\mathcal{L}_1$ adaptive control structure that interacted with AMPC via the desired closed-loop dynamics matrix $A_m$, reference $y_r$, and state $x(t)$.

The main results from the AMPC-$\mathcal{L}_1$ derivation are summarised below in algorithm form and may be interpreted as pseudocode. The full derivation and list of assumptions associated with AMPC-$\mathcal{L}_1$ are not repeated here, but may be found in [29]. Algorithm 1 describes the required computation for a single update of AMPC-$\mathcal{L}_1$ on the $k$-th iteration.

It was assumed that, prior to these computations, the system dynamics matrices were updated on the basis of the current flight condition.

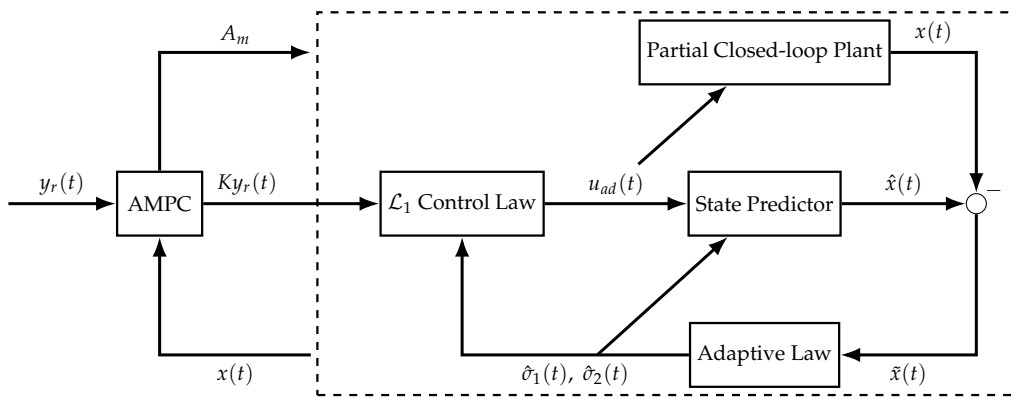

**Figure 1.** AMPC-$\mathcal{L}_1$ control structure for systems with uncertainties $\hat{\sigma}_1(t), \hat{\sigma}_2(t)$.

---

**Algorithm 1:** A single update of AMPC -$\mathcal{L}_1$ for the *k*-th iteration

---

**Data:** Reference command ($y_r$), current state ($x_k$), state prediction from previous update ($\hat{x}_k$), unmatched state from previous update ($x_{um,k}$)

**Result:** Optimal control input with uncertainty compensation $u_{\text{full}}$, updated state prediction ($\hat{x}_{k+1}$), updated unmatched state ($x_{um,k+1}$)

Compute AMPC optimal gain *K*:

$\quad \Phi(\delta t) = S \, \text{diag}(e^{\lambda_1(\delta t)}, \dots, e^{\lambda_k(\delta t)}) \, S^{-1}$
$\quad G \leftarrow CA^{-1}(\Phi(\delta t) - I)B_m$
$\quad F \leftarrow C\Phi(\delta t)$
$\quad K \leftarrow (G^T Q G + R)^{-1} G^T Q$

Compute desired closed-loop system dynamics:

$\quad A_m \leftarrow A - B_m K F$

Update estimated uncertainties ($\hat{\sigma}$) on the basis of the predicted state (from the last update):

$\quad B \leftarrow [B_m \; B_{um}]$
$\quad \Phi_{\text{ad}}(T_s) \triangleq A_m^{-1}(e^{A_m T_s} - \mathbb{I}_n),$
$\quad \tilde{x}_k \leftarrow \hat{x}_k - x_k, \text{ and}$
$\quad \mu_k \leftarrow e^{A_m T_s} \tilde{x}_k,$
$\quad \hat{\sigma}_{k,1} \leftarrow -\mathbb{I}_m B^{-1} \Phi_{\text{ad}}^{-1}(T_s) \mu_k$
$\quad \hat{\sigma}_{k,2} \leftarrow -\mathbb{I}_{n-m} B^{-1} \Phi_{\text{ad}}^{-1}(T_s) \mu_k$

Compute control input on the basis of the estimated uncertainties:

$\quad H_{m,\text{inv}} \leftarrow (1/ - (CA_m^{-1}B_m))$
$\quad \eta_1 \leftarrow \hat{\sigma}_{k,1}$
$\quad \eta_2 \leftarrow \hat{\sigma}_{k,2}$
$\quad \eta_{2m} \leftarrow H_{m,\text{inv}} C x_{um,k}$
$\quad u_{\text{raw}} \leftarrow \eta_1 + \eta_{2m} - K y_r$
$\quad u_{\text{ad}} \leftarrow \text{LowPassFilter}(-u_{\text{raw}}, \Delta t, \omega_k)$
$\quad u_{\text{full}} = u_{\text{ad}} - KF x_k$

Update the unmatched state:

$\quad \dot{x}_{um} \leftarrow A_m x_{um,k} + B_{um} \eta_2$
$\quad x_{um,k+1} \leftarrow x_{um,k} + \dot{x}_{um} \times \Delta t$

Compute state prediction for next update $\hat{x}_{k+1}$:

$\quad \dot{\hat{x}}_k \leftarrow A_m \hat{x}_k + B_m(u_{\text{ad}} + \hat{\sigma}_{k,1}) + B_{um} \hat{\sigma}_{k,2}$
$\quad \hat{x}_{k+1} \leftarrow \hat{x}_k + \dot{\hat{x}}_k \times \Delta t$

---

Matrix $G$ describes the forced response, $Q$ is the matrix of output error weights, and $R$ is a matrix of control input weights. $F$ is the free-response matrix with $F \in \mathbb{R}^{1 \times n}$, and $G \in \mathbb{R}$. $\delta t$ is the prediction horizon, and $A$, $B_m$, and $C$ are the state dynamics, control input, and measurement matrices, respectively, for a linear model as formulated below:

$$\dot{x} = Ax + B_m u \tag{14}$$

$$y = Cx \tag{15}$$

State transition matrix $\Phi(\delta t)$ was calculated using an eigendecomposition with $S$ that defined a set of eigenvectors of the system $\lambda_1$ to $\lambda_k$. $K$ is the optimal control matrix based on AMPC and was used to update the desired closed-loop dynamics ($A_m$) matrix, which is used by $\mathcal{L}_1$ adaptive control to predict the system response. The mismatch between the predicted and actual states was used to calculate the uncertainties ($\hat{\sigma}_{k,1}$, $\hat{\sigma}_{k,2}$). The calculated uncertainties were then used to compute the adaptive control input that compensated for the uncertainties.

*3.2. Longitudinal Control*

While the model used for the simulation and performance comparison in this paper was nonlinear, the longitudinal controller used the linearised angle of attack dynamics [30], which are as follows:

$$\dot{x}(t) = A(t)x(t) + B_m(t)u(t)$$

$$\begin{bmatrix} \dot{q}(t) \\ \dot{\alpha}(t) \end{bmatrix} = \begin{bmatrix} M_q(t) & M_\alpha(t) \\ \dot{\alpha}_q & -N_\alpha(t)/V(t) \end{bmatrix} \begin{bmatrix} q(t) \\ \alpha(t) \end{bmatrix} + \begin{bmatrix} M_{\delta_e}(t) \\ \frac{-N_{\delta_e}(t)}{V(t)} \end{bmatrix} u(t) \tag{16}$$

$$y(t) = Cx(t) = \begin{bmatrix} 0 & 1 \end{bmatrix} \begin{bmatrix} q(t) \\ \alpha(t) \end{bmatrix}, \tag{17}$$

where

$$M_\alpha = \frac{\bar{q}\bar{S}c}{I_2}C_{m_\alpha}, \quad N_\alpha = \frac{\bar{q}\bar{S}}{\bar{m}}C_{N_\alpha}, \quad M_q = \frac{\bar{q}\bar{S}c^2}{2I_2 V}C_{m_q}, \quad M_{\delta_e} = \frac{\bar{q}\bar{S}c}{I_2}C_{m_{\delta_e}},$$

$$N_{\delta_e} = \frac{\bar{q}\bar{S}}{\bar{m}}C_{N_{\delta_e}}.$$

Parameter $\dot{\alpha}_q$ is the pitch rate to the angle of the attack map. The linearisation of nonlinear models for aerospace vehicles is used to obtain linear time-variant models. Thus, system parameters are updated considering the current flight condition during each control update. The AMPC design parameters are shown in Table 1. $Q$ was chosen to prioritise performance. The same tuning parameters were used for all AMPC algorithms assessed in this paper.

**Table 1.** Design parameters of AMPC for longitudinal control.

| Parameter | Description | Value |
|:---:|:---:|:---:|
| Q | Weight for tracking error | 0.99 |
| R | Weight for control input | 0.001 |
| $\delta t_p$ | Prediction horizon | 0.5 s |

The design parameters of $\mathcal{L}_1$ adaptive control are shown in Table 2. The adaptive update time step is simply the control update rate for a control scheme that adapts at every control update. The low-pass filter cut-off frequency was selected to ensure the $\mathcal{L}_1$-norm condition [34].

**Table 2.** Design parameters of $\mathcal{L}_1$ adaptive control for longitudinal control.

| Parameter | Description | Value |
|-----------|-------------|-------|
| $\omega_c$ | Cut-off frequency of the low-pass filter | 20 Hz |
| $\delta t_{ad}$ | Time step for adaptive update | 0.002 s |

*3.3. Lateral Control*

The lateral control design used for AMPC and AMPC-$\mathcal{L}_1$ in this study utilises a model with combined roll and yaw dynamics with yaw rate $r$, side-slip angle $\beta$, roll rate $p$, and roll angle $\phi$ [30]. The linear state-space model for lateral dynamics is as follows:

$$\dot{x} = Ax + B_m u \tag{18}$$

$$\begin{bmatrix} \dot{r} \\ \dot{\beta} \\ \dot{p} \\ \dot{\phi} \end{bmatrix} = \begin{bmatrix} LN_r & LN_\beta & LN_p & 0 \\ -1 & \frac{Y_\beta}{V} & 0 & \frac{g}{V} \\ LL_r & LL_\beta & LL_p & 0 \\ 0 & 0 & 1 & 0 \end{bmatrix} \begin{bmatrix} r \\ \beta \\ p \\ \phi \end{bmatrix} + \begin{bmatrix} 0 & LN_{\delta_r} \\ 0 & Y_{\delta_r}/V \\ LL_{\delta_a} & 0 \\ 0 & 0 \end{bmatrix} \begin{bmatrix} \delta_a \\ \delta_r \end{bmatrix} \tag{19}$$

$$y = Cx = \begin{bmatrix} 1 & 0 & 0 & 0 \\ 0 & 1 & 0 & 0 \\ 0 & 0 & 0 & 1 \end{bmatrix} \begin{bmatrix} r \\ \beta \\ p \\ \phi \end{bmatrix}, \tag{20}$$

where

$$LN_\beta = \frac{\bar{q}\bar{S}b}{I_3}C_{n_\beta}, \quad LN_r = \frac{\bar{q}\bar{S}b^2}{2I_3V}C_{n_r}, \quad LN_p = \frac{\bar{q}\bar{S}b^2}{2I_3V}C_{n_p}, \quad Y_\beta = \frac{\bar{q}\bar{S}}{\bar{m}}C_{y_\beta},$$

$$LL_r = \frac{\bar{q}\bar{S}b^2}{2I_1V}C_{l_r}, \quad LL_p = \frac{\bar{q}\bar{S}b^2}{2I_1V}C_{l_p}, \quad LL_\beta = \frac{\bar{q}\bar{S}b}{I_1}C_{l_\beta}.$$

Algorithm 1 was applied generically to both longitudinal and lateral dynamics. The design parameters for lateral control using AMPC-$\mathcal{L}_1$ are shown in Tables 3 and 4. The prediction time horizon was chosen to balance the need for aggressive short- and long-term corrections.

**Table 3.** Design parameters of AMPC for lateral control.

| Parameter | Description | Value |
|-----------|-------------|-------|
| $Q_\phi$ | Tracking error penalty weight for roll angle | 0.99 |
| $Q_\beta$ | Tracking error penalty weight for side-slip angle | 0.9 |
| $Q_r$ | Tracking error penalty weight for yaw rate | 0.5 |
| $R_{\delta_a}$ | Control input penalty weight for aileron | 0.001 |
| $R_{\delta_r}$ | Control input penalty weight for rudder | 0.001 |
| $\delta t_p$ | Prediction horizon | 0.5 s |

**Table 4.** Design parameters of $\mathcal{L}_1$ adaptive control for lateral control.

| Parameter | Description | Value |
|-----------|-------------|-------|
| $\omega_c$ | Cutoff frequency for the low pass filter | 20 Hz |
| $\delta t_{ad}$ | Time step for adaptive update | 0.002 s |

## 4. Results and Discussion

Three different control algorithms are assessed for their control performance under various simulated conditions in this section. The control algorithms were pole placement [33], baseline AMPC, and AMPC-$\mathcal{L}_1$ [29].

At the start of stage separation, the booster was assumed to have an initial attitude of $\phi = 90°$, $\theta = 0°$, and $\psi = 0°$. First, the booster must undergo a stage separation maneuver to detach itself from the launch stack quickly and safely. Then, it performs a reorientation to an attitude of $\phi = 90°$. Lastly, it decelerates in a controlled manner, performing a pull-up maneuver to further decrease its speed. In a practical scenario, the vehicle then deploys wings for a fly-back phase, landing at the launch site. However, the final fly-back phase was not considered in this analysis. Figure 2 shows the nominal commands assumed to come from the guidance law for stage separation, reorientation, and descent.

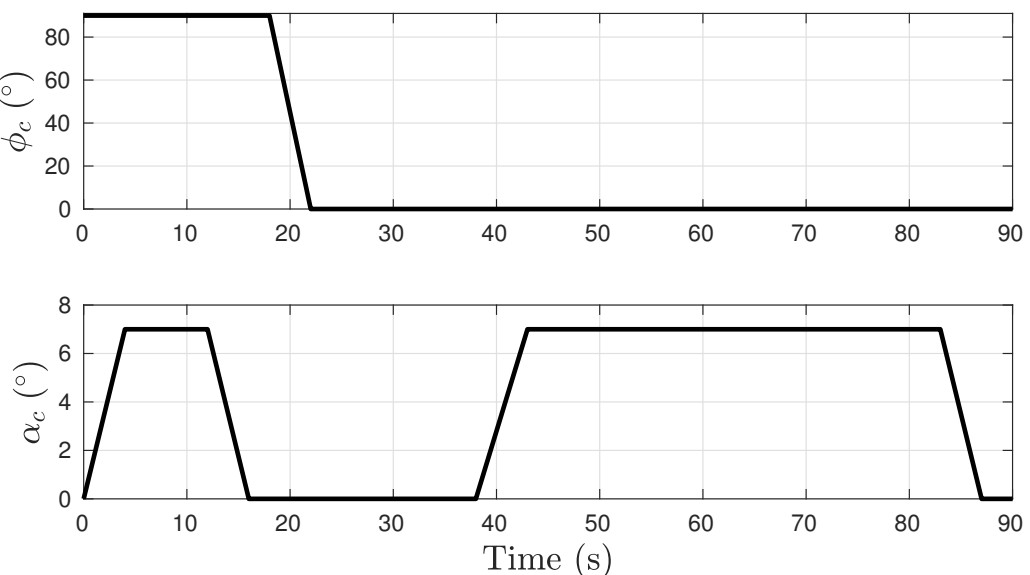

**Figure 2.** Nominal guidance commands for a fly-back booster during re-entry.

The stage separation phase lasts from $t = 0$ to $t = 18$ seconds. This is marked by a ramp-up of commanded $\alpha$ while keeping $\phi = 90°$. The reorientation phase involves ramping the commanded roll to zero while keeping the $\alpha = 0$. The pull-up consists of ramping the commanded $\alpha$ to $7°$. For the AMPC controllers, side slip ($\beta$) and yaw rate ($r$) are regulated, but only the yaw rate is regulated for the pole placement.

### 4.1. Nominal Case

In the nominal case, the control algorithms were assumed to have perfect knowledge of the dynamic model with a deterministic set of guidance commands. The guidance and control loops were updated at 500 Hz. Figures 3–6 show the control performance of the pole placement, AMPC, and AMPC-$\mathcal{L}_1$ under nominal conditions.

Figure 3 shows that all controllers had similar levels of performance, with AMPC-$\mathcal{L}_1$ exhibiting slightly less error during the ramping up and down. This was due to the ability of the $\mathcal{L}_1$ adaptations to compensate for the model mismatch arising from the nonlinearities.

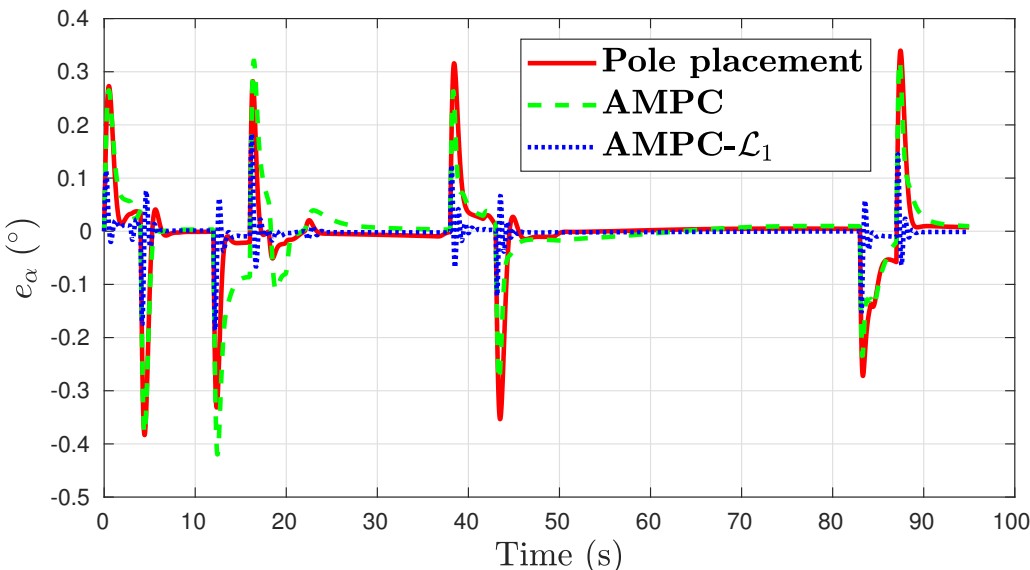

**Figure 3.** Angle-of-attack tracking error for nominal case.

Figure 4 shows that the baseline AMPC had the worst performance, accumulating errors of over 20° during the reorientation maneuver before converging back to zero during the pull-up maneuver. The pole placement controller resulted in steady-state error throughout the stage separation phase. Arguably, the AMPC and pole placement controllers may be tuned further to produce better error tracking. However, the AMPC-$\mathcal{L}_1$ controller, which had the same tuning parameters as those of the baseline AMPC, could eliminate the steady-state error and keep the tracking error well below 5°.

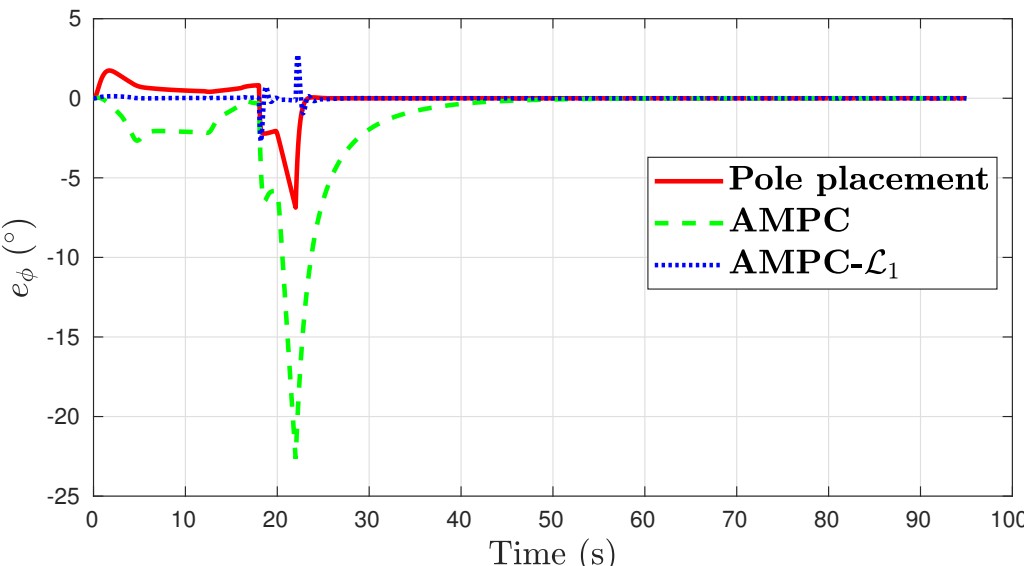

**Figure 4.** Roll tracking error for the nominal case.

The side-slip regulation performance is shown in Figure 5. The stage-separation and reorientation maneuvers produced perturbations in lateral velocity, which translated into disturbances to the side-slip angle. All controllers performed acceptably, keeping the tracking error below 2°.

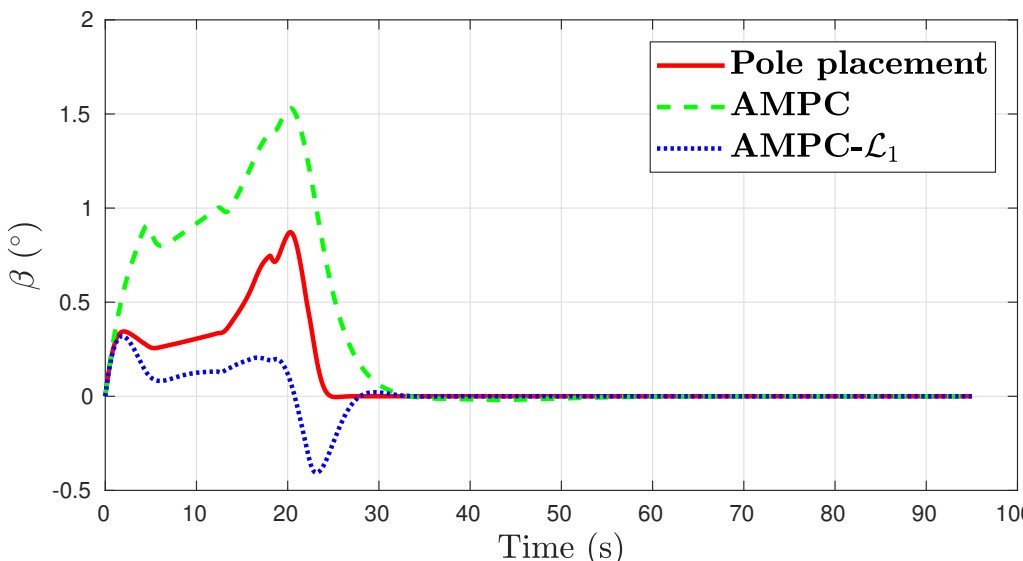

**Figure 5.** Sideslip regulation error for the nominal case.

The yaw-rate regulation performance is shown in Figure 6. The stage-separation and reorientation maneuvers produced perturbations in the yaw rate. All controllers performed acceptably, keeping the tracking error below $1°/s$.

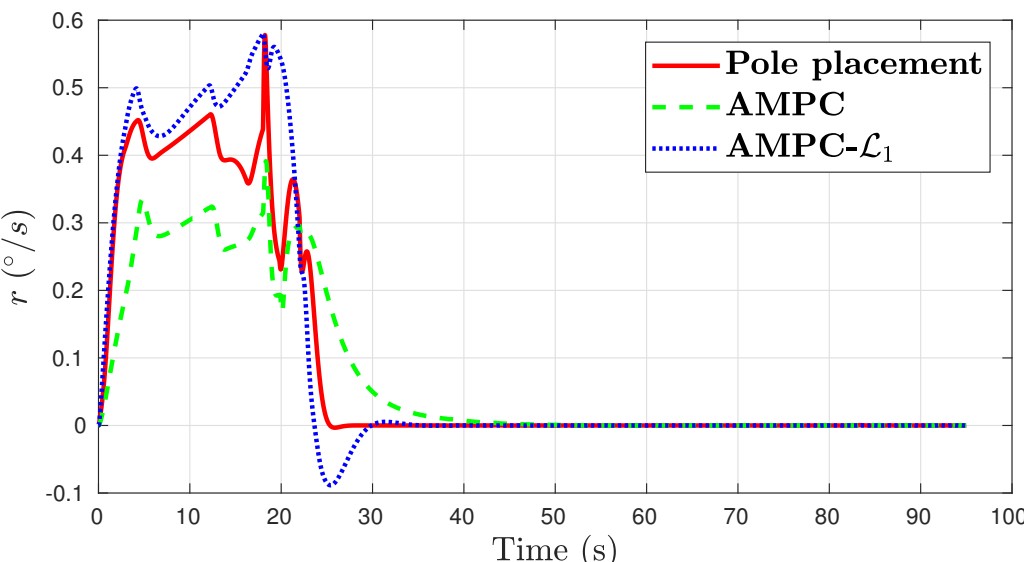

**Figure 6.** Yaw-rate regulation error for the nominal case.

*4.2. Aerodynamic Uncertainties*

Although the control algorithms performed acceptably under nominal conditions, it is important to consider the effects of aerodynamic uncertainty on the control performance. In reality, the controller always has imperfect knowledge of the aerodynamic parameters. The control performance of the pole placement, AMPC, and AMPC-$\mathcal{L}_1$ was assessed in the presence of an aerodynamic model mismatch. Aerodynamic errors were introduced into the nonlinear simulation model on the basis of past research pertaining to the relative accuracy of modern aerodynamic prediction methods for re-entry vehicles [35]. The aerodynamic errors are shown in Table 5.

**Table 5.** Aerodynamic errors injected into the simulation model [35].

| Aerodynamic Parameter | % Error |
|---|---|
| $C_L$ | 21 |
| $C_D$ | 53 |
| $C_m$ | −34 |
| $C_{m_{\delta e}}$ | 46 |
| $C_{m_q}$ | −66 |
| $C_{l_\beta}$ | 4.4 |
| $C_{n_\beta}$ | 7.3 |
| $C_{y_\beta}$ | 12.7 |
| $C_{l_{\delta a}}$ | 5.57 |
| $C_{y_{\delta a}}$ | −4.52 |
| $C_{n_{\delta r}}$ | −6.9 |
| $C_{y_{\delta r}}$ | 17 |
| $C_{l_p}$ | −2.85 |
| $C_{l_r}$ | −80.4 |
| $C_{n_p}$ | 50.4 |
| $C_{n_r}$ | 23.77 |

The aerodynamic errors were not intended to capture the worst-case scenario; rather, they represent a realistic case that is within the bounds of possible errors. The simulations were run with the aerodynamic errors, and the results are shown in Figures 7–10.

Figure 7 shows that the pole placement controller exhibited large transient and steady-state errors. The transient error occurred during stage separation and pull-up maneuvers. Both AMPC and AMPC-$\mathcal{L}_1$ exhibited acceptable control performance, while AMPC-$\mathcal{L}_1$ had the lowest tracking error.

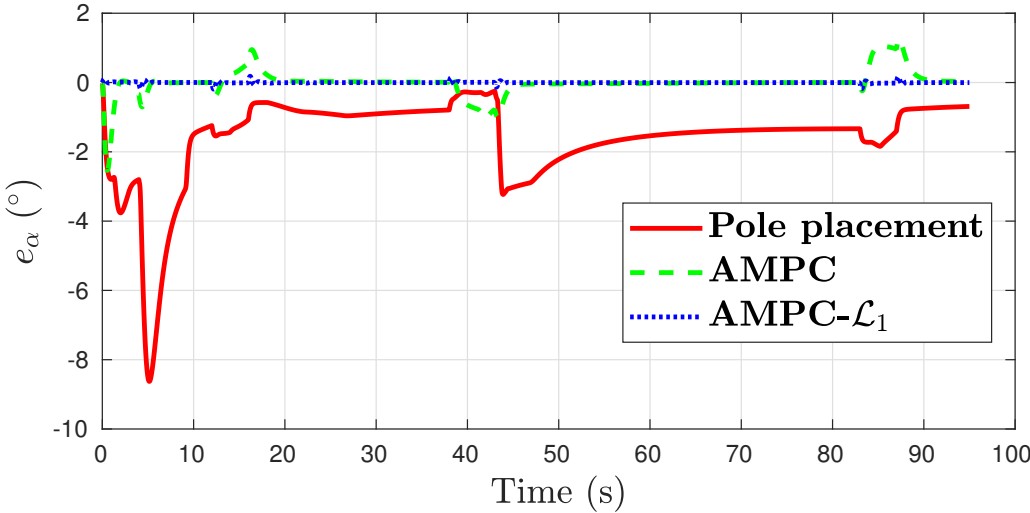

**Figure 7.** Angle-of-attack tracking error for aerodynamic uncertainty case.

In the roll tracking task, a large transient error of over 50° was present when baseline AMPC was used, as shown in Figure 8. Compared to the results from the nominal case in Figure 4, the pole placement and AMPC controllers produced double the maximal transient error in the presence of aerodynamic errors. This is in contrast to AMPC-$\mathcal{L}_1$, which retained the control performance of the nominal case.

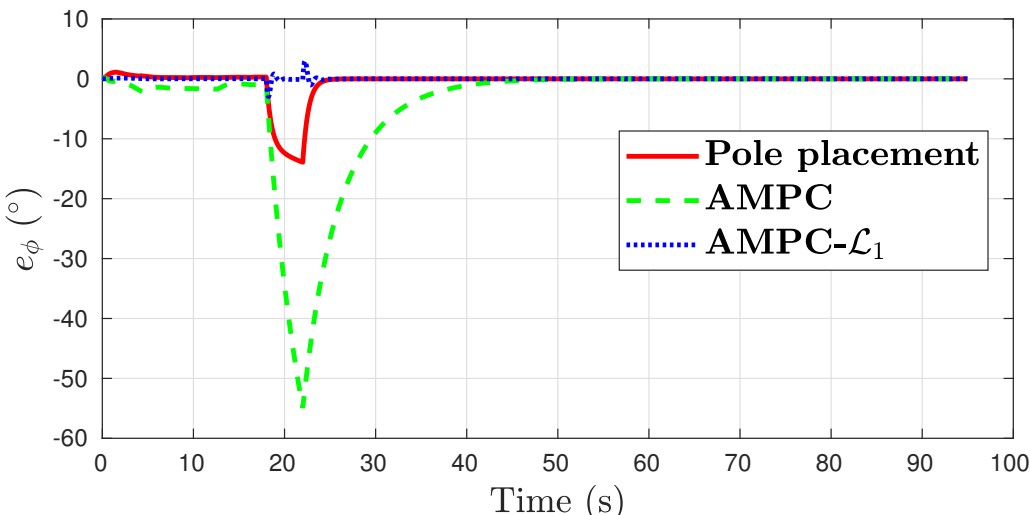

**Figure 8.** Roll tracking error for aerodynamic uncertainty case.

Compared to the nominal case in Figure 5, all controllers produced similar levels of side-slip performance, as shown in Figure 9.

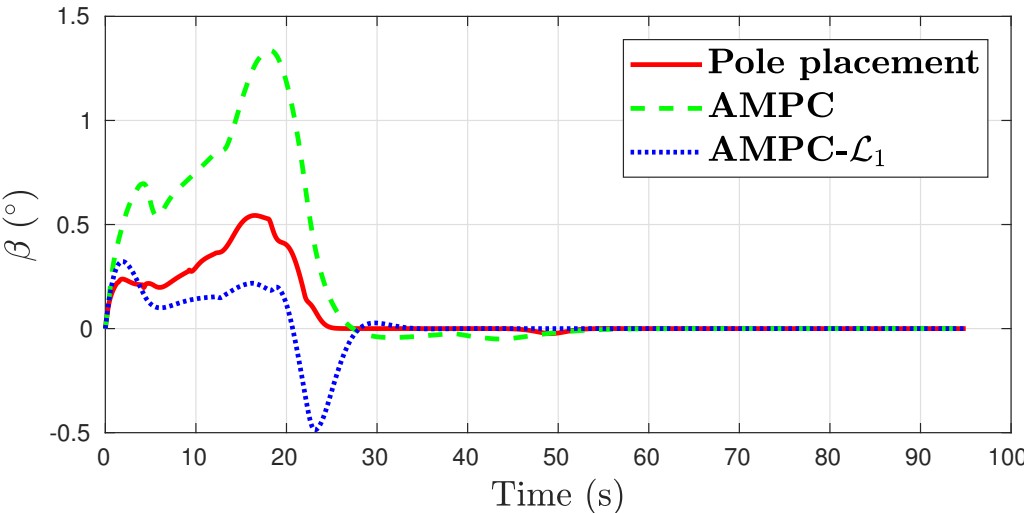

**Figure 9.** Sideslip regulation error for aerodynamic uncertainty case.

Compared to the nominal case in Figure 6, the pole placement and AMPC-$\mathcal{L}_1$ controllers achieved similar performance in yaw-rate regulation as shown in Figure 10. However, the performance of the baseline AMPC slightly deteriorated.

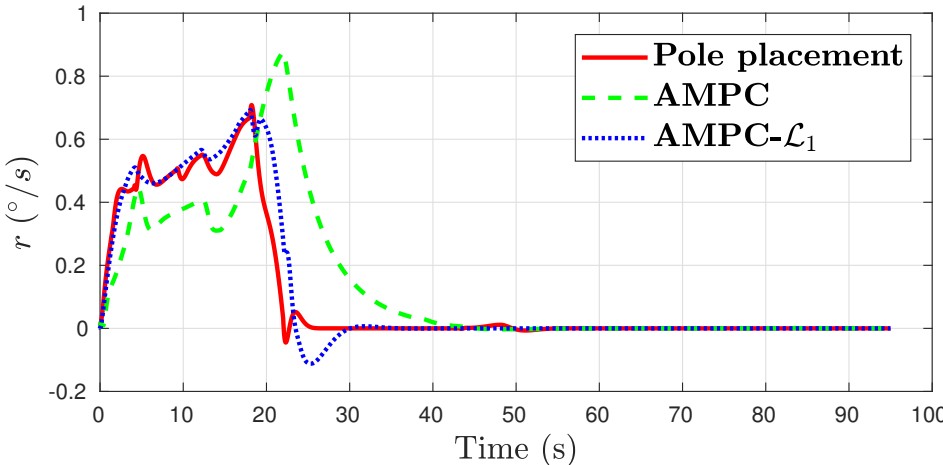

**Figure 10.** Yaw rate regulation error for aerodynamic uncertainty case.

*4.3. Guidance Law Uncertainties*

The focus of this study was evaluating the attitude control performance of pole placement, AMPC, and AMPC-$\mathcal{L}_1$. So far, the guidance law was assumed to produce a certain sequence of $\alpha$ and $\phi$ commands that were sent to the attitude controllers. However, in reality, the guidance law might produce different commands to what was originally planned, reacting to the vehicle's current state to fulfil a broader mission plan. For instance, $\alpha$ commands may be increased or decreased in order to change the vehicle drag, thereby changing its deceleration. The $\phi$ commands may also be increased or decreased during the pull-up phase to allow the vehicle to track a certain heading.

The specifics of the guidance law design and requirements are outside the scope of this study. However, the uncertainty of the reference commands tracked by the attitude controllers may be considered. Therefore, a change in guidance commands is considered in this section. There were no further changes to the controller design parameters (Figure 11).

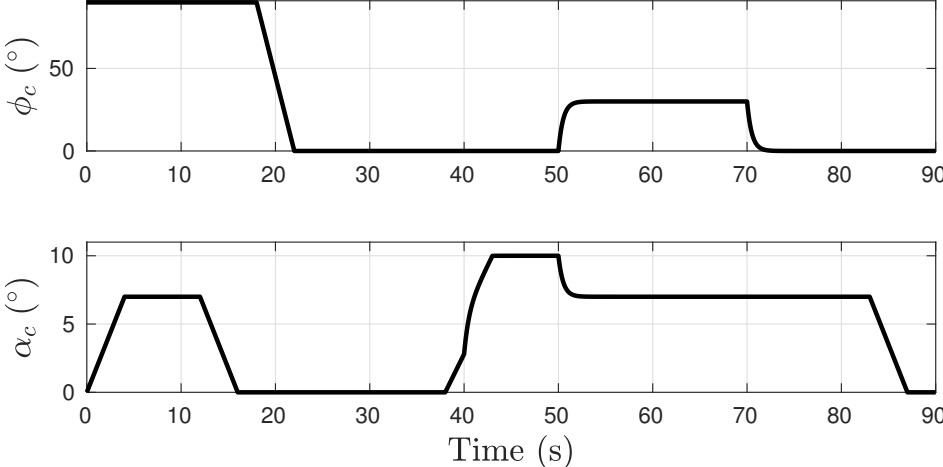

**Figure 11.** Modified guidance commands for a fly-back booster during re-entry.

The control performance plots are shown in Figures 12–15. The three control algorithms show a similar $\alpha$ tracking performance. Baseline AMPC has large roll tracking error of over 20°. The pole placement yaw damper results in the largest error in side slip and yaw rate. In all channels, the AMPC-$\mathcal{L}_1$ achieves the best control performance.

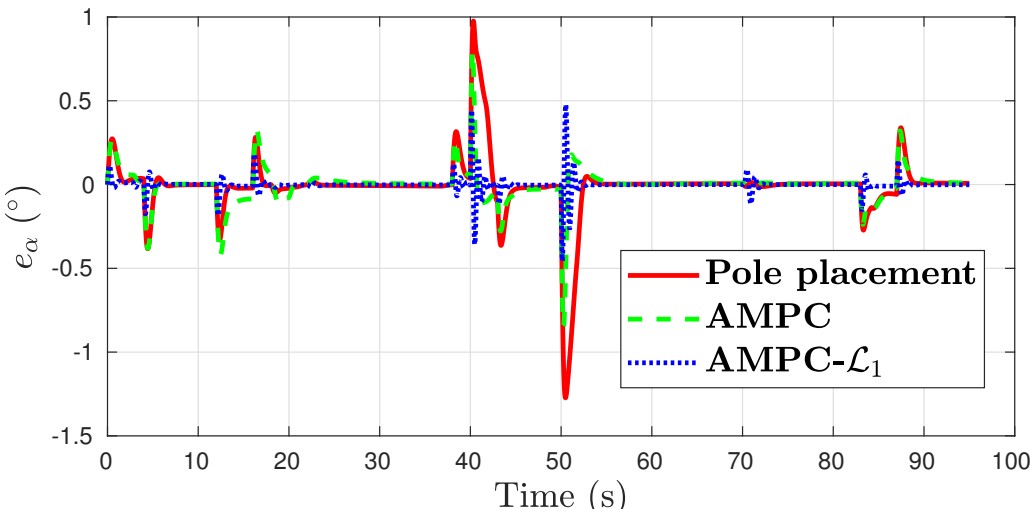

**Figure 12.** Angle-of-attack tracking error for guidance uncertainty case.

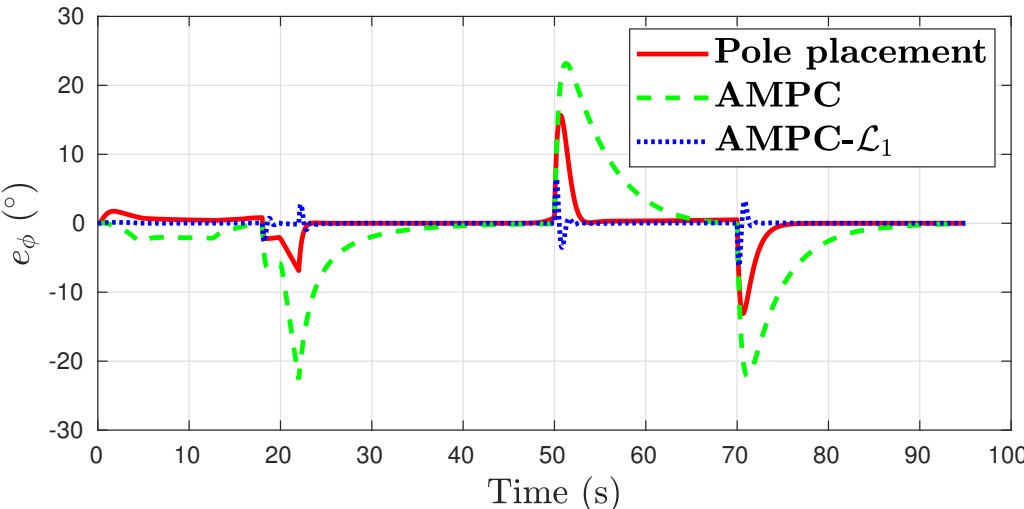

**Figure 13.** Roll tracking error for guidance uncertainty case.

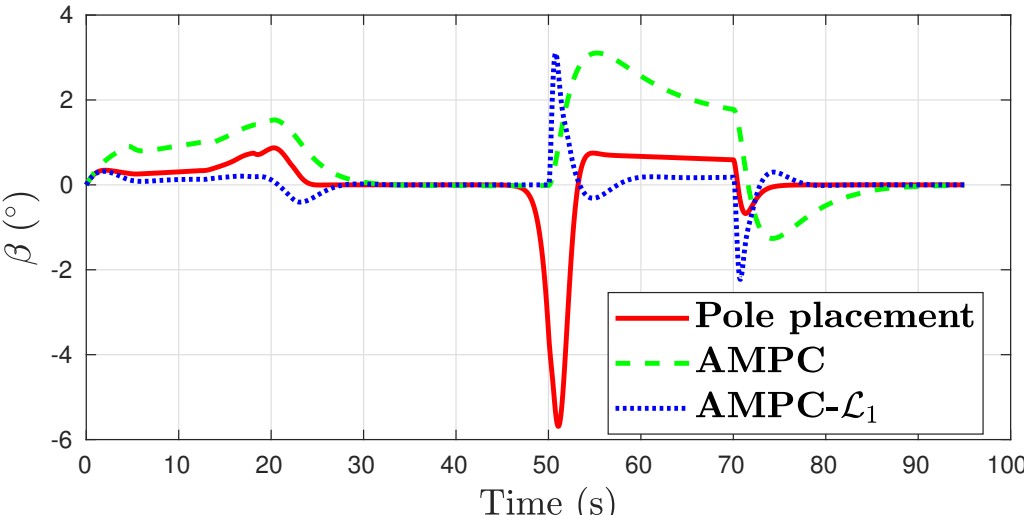

**Figure 14.** Sideslip regulation error for guidance uncertainty case.

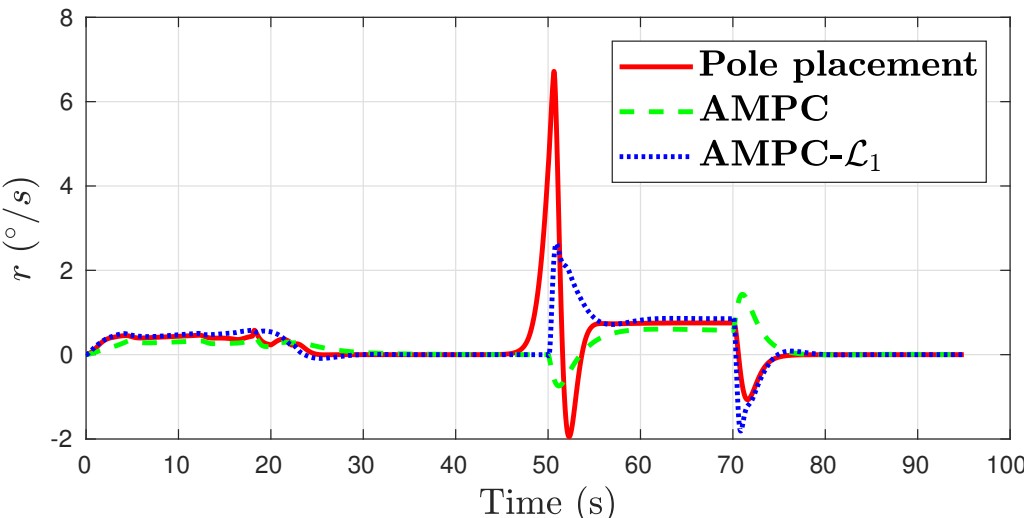

**Figure 15.** Yaw rate regulation error for guidance uncertainty case.

*4.4. Computational Expense Characterisation*

This section characterises the computational expense of running AMPC-$\mathcal{L}_1$ on an embedded Linux computer. A hardware-in-the-loop simulation is configured to examine the computational loads. The main goal of the study presented in this section is to establish a set of feasible loop rates given the software, algorithm, and hardware implementation for both AMPC and AMPC-$\mathcal{L}_1$. The embedded computer is Raspberry Pi 3 Model B with a 1.2G Hz 64-bit CPU and 1 GB RAM. Modern flight computers typically have better processing and RAM than those, rendering this a conservative test.

The hardware-in-the-loop simulation in this study used the models and controller described in Section 3.3. The lateral dynamics with four states and two control inputs are simulated with the AMPC and AMPC-$\mathcal{L}_1$ control schemes, implemented using Algorithm 1. Two software applications are required: a dynamics simulator that runs on a native x86 computer and the flight controller application that runs on the Raspberry Pi.

The communications between the simulator and the flight controller follow a server-client model using TCP sockets. However, the `send()` and `recv()` methods are repeated synchronously until the end of the simulation. The simulator application:

1. Receives control input data from the flight controller application.
2. Processes the dynamics and updates the state of the vehicle.
3. Sends the state of the vehicle back to the flight controller application.

The flight controller application similarly performs the following tasks:

1. Receives the state of the vehicle from the simulator application.
2. Computes the best control inputs on the basis of vehicle state.
3. Sends the computed control input data back to the simulator application.

The x86 computer used to run the simulator application was an Intel Core i7-8550U CPU @ 1.80 GHz with 8 GB RAM. Due to the much faster processor on the x86 computer, any bottlenecks in computational performance may be attributed to the flight controller application. Both applications are single-threaded.

A network was set up with Raspberry Pi and the x86 computer. The network router was isolated from the Internet and any other network traffic in order to create a deterministic and fair testing environment. As such, only the two above-mentioned devices were configured on the network.

The software toolchain used to build the simulator and flight controller applications is presented in Table 6.

**Table 6.** Toolchain for building flight controller and simulator applications.

|  | **Simulator** | **Flight Controller** |
|---|---|---|
| C Compiler | `GCC` 7.5.0 | `arm-linux-gnueabihf-gcc` 8.3.0 |
| C++ Compiler | `G++` 7.5.0 | `arm-linux-gnueabihf-g++` 8.3.0 |
| Runtime Optimisation | `-O3` | `-O3` |
| Eigen3 | 3.3.4 | 3.3.4 |
| cmake | 3.14.4 | 3.14.4 |
| C++ | 17 | 17 |

Both applications were built on Ubuntu 20. The optimisation compiler option was set to prioritise run-time performance at the expense of compilation time and executable size.

A Linux programme `htop` was used to record the CPU % usage. The CPU usage in one simulation may be averaged to obtain a performance metric for comparing the computational load at different loop update rates. These results are presented in Table 7. On the basis of the results in Table 7, even at a 1 KHz loop rate, the average CPU usage of a single core on the Raspberry Pi 3 was clearly only 39% for AMPC-$\mathcal{L}_1$. The $\mathcal{L}_1$ augmentation increased the average CPU load by approximately 3%, less than 10% of the computational usage due to baseline AMPC, demonstrating the suitability of AMPC-$\mathcal{L}_1$ as an adaptive and efficient model predictive control algorithm for implementation on embedded systems.

**Table 7.** Comparison of computational load on the Raspberry Pi at various loop update rates.

| **Control Scheme** | **Loop Rate (Hz)** | **CPU %** |
|---|---|---|
| AMPC | 100 | 7% |
| AMPC-$\mathcal{L}_1$ | 100 | 7.5% |
| AMPC | 200 | 14.5% |
| AMPC-$\mathcal{L}_1$ | 200 | 14.5% |
| AMPC | 333 | 19.2% |
| AMPC-$\mathcal{L}_1$ | 333 | 20.3% |
| AMPC | 500 | 22.5% |
| AMPC-$\mathcal{L}_1$ | 500 | 22.9% |
| AMPC | 1000 | 36% |
| AMPC-$\mathcal{L}_1$ | 1000 | 39% |

## 5. Conclusions

In this paper, the AMPC-$\mathcal{L}_1$ adaptive model predictive control algorithm was implemented on a nonlinear simulation of fly-back booster stage separation and re-entry. The performance of AMPC-$\mathcal{L}_1$ was evaluated against baseline AMPC and pole placement controllers under nominal conditions, aerodynamic uncertainties, and guidance law uncertainties. The controllers are synthesised using linear models for longitudinal and lateral dynamics, which are successively linearised on the basis of the latest flight condition. In all cases, AMPC-$\mathcal{L}_1$ showed superior control performance compared to baseline AMPC and pole placement. AMPC-$\mathcal{L}_1$ could compensate for model mismatch due to nonlinearities and other sources of uncertainty to preserve the nominal performance of AMPC. The computational load usage of AMPC-$\mathcal{L}_1$ was assessed and benchmarked against the baseline AMPC. AMPC-$\mathcal{L}_1$ was capable of being updated up to 1 KHz on a single core of a Raspberry Pi Model 3 B while utilising an average CPU load of 39%, which demonstrates the suitability of AMPC-$\mathcal{L}_1$ for implementation on resource-constrained systems.

**Author Contributions:** Conceptualisation, J.C. and E.K.; methodology, J.C. and E.K.; software, J.C.; validation, J.C.; investigation, J.C. and E.K.; writing—original draft preparation, J.C.; writing—review and editing, E.K.; supervision, E.K. All authors have read and agreed to the published version of the manuscript.

**Funding:** The second author was a recipient of the Australian Government Research Training Program stipend while undertaking the research activities that culminated in the publication of this note.

**Data Availability Statement:** Data available on request from the authors.

**Conflicts of Interest:** The authors declare no conflict of interest.

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
