# Peer review of "Nonlinear Simulation and Performance Characterisation of an Adaptive Model Predictive Control Method for Booster Separation and Re-Entry"

_electronics, doi:10.3390/electronics12061488_

Round 1

Reviewer 1 Report (New Reviewer)

Generally, this paper is interesting and can be accepted by carefully considering the following concerns. (1) The contributions can hardly be observed from the abstract; (2) Literature review should be enhanced by covering recently related papers, such as 10.1109/TVT.2021.3136670, 10.1109/TCST.2019.2955657; (3) Stability analysis should be completely provided to ensure that the entire closed-loop control system is stable; (4) The Algorithm 1 can be made much more logic and easy to follow for readers; (5) Analysis on simulation results can be arranged in a much clearer manner.

Author Response

Thanks to Reviewer 1 for their comments. 

1) The contribution of the paper is to evaluate the L_1 adaptive model predictive control (AMPC-L_1) method in terms of its control performance and computational load. This has been colored in the new version of the paper.

2) The suggested references are unrelated to adaptive control, learning-based control, model predictive control, or fly-back booster vehicles. We could not see why these papers should be added. 

3) The stability of the algorithm has been given in Ref. (29).

Chai, J.; Medagoda, E.; Kayacan, E. Adaptive and Efficient Model Predictive Control for Booster Reentry. Journal of Guidance, Control, and Dynamics 2020, 43, 2372–23. 

4) Algorithm 1 is made easy to follow. 

5) Analysis of simulation results is arranged again. 

Reviewer 2 Report (New Reviewer)

The article topic is interesting and useful. In my view the contribution needs significant work before it can be considered for publication.

In general, the lack of any details on the pole placement solution for the considered problem is a serious omission for the article. Page 6 provides enough free space for this.

Additionally, the article itself is rather chaotic. If this is meant to be a follow-up article, this should be cleanly stated and the link between the previous and current work should be vastly strengthened by (when appropriate) citing the formulas/theorem numbers from the original (main) article.
As it stands, it is very difficult to judge the quality of the work. Another problem is that the crucial AMPC algorithm's details of reference [29] are hidden behind a paywall which adds to the difficulty of rewiewing the presented work.

Additionally, I managed to find a number of less important details outlined below:
In (3), what are I_{a,b}?
page 3 - you state that the goal of your contribution is to: "(iii) describ[e] the implementation of AMPC-L1 on an embedded Linux computer". Where exactly is this description given? From an implementation perspective, lines 271-301 only show rather simple network-based programming while giving no details on actual implementation of the Algorithm 1 for the Raspberry Pi.
page 5 - is "G" a matrix or a scalar?
page 5 - in (14) did you mean "B_m" as the sentence in line 144 would suggest?
page 5 - what is \bar S below (17)?
page 6, in Algorithm 1: what is "I" compared to \mathbb{I}?
page 7 - in (18)-(20) maybe consider changing the variable names to avoid confusion with (16)-(17)?
page 11 - what is the rationale behind Table 5 as those values seem rather precise?
page 14 - what operating system was the Raspberry pi using?

In general, the article lacks a well-stated goal. It contains:
1) no new theoretical work,
2) it does not give any useful tips on implementing a known algorithm,
3) it lacks too many details to allow anyone to even try to repeat the outlined results. Those details are hidden behind a paywall which requires a significant effort to obtain them.
If at least one of 1), 2), or 3) were done, the article would be worthy of publication. As it stands, however, its usefulness is too limited to be interesting to a broader audience.

As such, I suggest the Authors to resubmit it after significant re-structuring of the work.

Author Response

Thanks to Reviewer 2 for their nice comments. 

Comment 1: Additionally, the article itself is rather chaotic. If this is meant to be a follow-up article, this should be cleanly stated and the link between the previous and current work should be vastly strengthened by (when appropriate) citing the formulas/theorem numbers from the original (main) article.
As it stands, it is very difficult to judge the quality of the work. Another problem is that the crucial AMPC algorithm's details of reference [29] are hidden behind a paywall which adds to the difficulty of rewiewing the presented work.

Reply 1: Thanks to Reviewer 2 for this comment. The paper has been added to arxiv.org. Therefore, it is not a hidden behind a paywall reference. 

Comment 2: In (3), what are I_{a,b}? 

Reply 2: l_{11}, l_{12}, etc are the inertia parameters that depend on the system. 

Comment 3: page 3 - you state that the goal of your contribution is to: "(iii) describ[e] the implementation of AMPC-L1 on an embedded Linux computer". Where exactly is this description given? From an implementation perspective, lines 271-301 only show rather simple network-based programming while giving no details on actual implementation of the Algorithm 1 for the Raspberry Pi. 

Reply 3: Thanks to Reviewer 2 for this comment. The outline is given in the paper. If we give the actual commands, then it would be like a lecture note. 

Comment 4: page 5 - is "G" a matrix or a scalar?

Reply 4: G is a matrix (equal to ?? (Φ(??)−?) ?_m). It is given in Algorithm 1 . This has been clarified in the new version of the paper. 

Comment 5: page 5 - in (14) did you mean "B_m" as the sentence in line 144 would suggest? 

Reply 5: Thanks to Reviewer 2 for this comment. It is B_{m}. This has been corrected in the new version of the paper. 

Comment 6: page 5 - what is \bar S below (17)?

Reply 6: It is a system parameter given in Ref [30], which is cited in the paper.

Comment 7: page 6, in Algorithm 1: what is "I" compared to \mathbb{I}? 

Reply 7: l is a system parameter \mathbb(I) is an identity matrix. 

Comment 8: page 7 - in (18)-(20) maybe consider changing the variable names to avoid confusion with (16)-(17)?

Reply 8: \dot{x} = Ax + Bu is a standard state-space formulation. It is not clear what the Reviewer 2 suggested. 

Comment 9: page 11 - what is the rationale behind Table 5 as those values seem rather precise?

Reply 9: Aerodynamic errors are taken from Ref [35]. The paper has been cited. 

Comment 10: page 14 - what operating system was the Raspberry pi using?

Reply 10: It is a raspberry Pi so, it is Raspberry Pi operating system. 

Round 2

Reviewer 2 Report (New Reviewer)

The article after the introduced changes is vastly improved.
There is one critically important part that needs to be addressed before publication: some way of adding the link "https://arxiv.org/pdf/2103.12536.pdf" to the reference [29] must be included into the paper, otherwise it will still be too time-consuming to find.  

Authors should also verify that all variables in (1)-(3) are explained in the text. A simple information, like in Reply 2 ("l_{11}, l_{12}, etc are the inertia parameters") is, in my view, enough. The same applies to Replies 6 and 7.

Remark 8/Reply 8 - please disregard this comment as this was a result of me misreading part of the article.

Reply 9 - Table 5 heading should also include the reference to [35]. The same applies to all other tables that use values from other works, if applicable.

Reply 10. From e.g. https://www.javatpoint.com/best-operating-system-for-raspberry-pi
there are about 20 different operating systems that can be installed on Raspberry Pi. Therefore I still think my suggestion to add the information that Raspberry PI was using the Raspberry Pi OS holds.

Author Response

Comment 1: The article after the introduced changes is vastly improved.

Response 1: Thanks to Reviewer for his kind comments. 

Comment 2: There is one critically important part that needs to be addressed before publication: some way of adding the link "https://arxiv.org/pdf/2103.12536.pdf" to the reference [29] must be included into the paper, otherwise it will still be too time-consuming to find.  

Response 2: The link has been added to the reference. 

Comment 3: Authors should also verify that all variables in (1)-(3) are explained in the text. A simple information, like in Reply 2 ("l_{11}, l_{12}, etc are the inertia parameters") is, in my view, enough. The same applies to Replies 6 and 7.

Response 3: The following part has been added in red to the paper:

"$I$ is the inertia matrix, $m$ is the mass, $g$ is the gravitational force, $f_{a}$ is the aerodynamic forces, $m_{a}$ is the aerodynamic moments"

Please page 3. 

Comment 4: Table 5 heading should also include the reference to [35]. The same applies to all other tables that use values from other works, if applicable.

Response 4 The reference [34] has been cited in the Table 5 heading. 

Comment 5:From e.g. https://www.javatpoint.com/best-operating-system-for-raspberry-pi
there are about 20 different operating systems that can be installed on Raspberry Pi. Therefore I still think my suggestion to add the information that Raspberry PI was using the Raspberry Pi OS holds.

Response 5: Thanks to Reviewer for this comment. The OS is Ubuntu. This has been added to the paper in red.

This manuscript is a resubmission of an earlier submission. The following is a list of the peer review reports and author responses from that submission.

Round 1

Reviewer 1 Report

1. The paper deals with the implementation and comparison of a control algorithm in a booster separation and reentry task. Basically, the work is a kind of report that presents known control techniques used for a particular non-linear process. For this reason, no new theoretical results can be seen and Section 3 makes use of descriptions presented in the literature. However, it should be emphasised that the authors honestly give references to the sources they use.

2. It is somewhat disappointing to use approximations of linear non-linear processes for algorithm synthesis. I think that referring to the original nonlinear models could give some new theoretical result and provide material for comparisons that would assess the usefulness of such an approach.

3. In my opinion, the main contribution of the authors is the implementation of the algorithm using an embedded system and the simulation tests performed. Unfortunately, there is no discussion about the stability of the proposed solution. The authors do not state when the results obtained are no longer acceptable and what this depends on. Finally, the comparisons of the simpler algorithms do not take into account the control cost and only present the error waveforms. Such comparisons are not objective.

4. Implementation results do not show what the distribution of computation times is and whether the real-time operation is maintained. I believe this is a critical issue when highlighting the results of a controller implementation that is supposed to stabilise a closed-loop process. I would see a more robust discussion and presentation of detailed results here, not just CPU load.

5. Minor note: there are incorrect references to Algorithm 1.

Author Response

Thanks to Reviewer 1 for their comments. Please kindly find our responses to their comments.

1)  This paper extends our previous paper by: (i) including stage separation in the mission scenario; (ii) using a high-fidelity nonlinear simulation model with both longitudinal and lateral dynamics; and (iii) describing the implementation of AMPC-L1 on an embedded Linux computer. 

2) Comment is not clear to us. 

3) The stability of L_{1} adaptive control method with MPC discussed in literature. We did not need to discuss this in the paper as the scope of the paper is "Adaptive Model Predictive Control Method for Booster Separation and Reentry". 

4) As the algorithm solves an optimization problem for model predictive control, for sure most of the computation power is for solving the MPC problem. 

5) Please let us know the incorrect references. 

Reviewer 2 Report

The paper has old references and the latest reference is for 2020, between which are the papers published with the same group of authors.

The main novelty of the present work with the paper  28, 29, 33, 32 and other previous works of the authors should be clarified. The authors claim that detailing the implementation process is new in this paper however, I can see the section B and D in the previous paper of the same authors with similar descriptions. please clarify it. 

The authors claim that (In all cases, AMPC-L1 shows to exhibit superior control performance compared to 308 baseline AMPC and pole placement. ) - (Results show that AMPC-L1 exhibits superior control performance under nominal 5 conditions as well as under aerodynamic and guidance law uncertainties. ) which is not seen from the simulations in figure 6,9,10,15, The authors should conclude their claims by the simulations not by their personal belief. 

Author Response

Thanks To Reviewer 2 for their comments. Please kindly find our responses to their comments below:

This paper extends the existing research by: (i) including stage separation in the mission scenario; (ii) using a high-fidelity nonlinear simulation model with both longitudinal and lateral dynamics; and (iii) describing the implementation of AMPC-L1 on an embedded Linux computer.

The controllers control the longitudinal and lateral dynamics. So, we control \phi and \alpha angles. if Reviewer 2 checks Figures 3, 4, 7, 8, 12, 13, etc., AMPC-L1 exhibits superior control performance. We gave other results to show how other angles are changing by time. They do not need to be better as they are not controlled. 

Round 2

Reviewer 1 Report

I believe that the problem considered is of significant practical importance, however, the manuscript does not present a new solution. In my opinion, it is more suitable as a conference paper than for a JCR-listed journal paper.

Regarding my comment 2, I wrote earlier that the controller synthesis is based on linear approximation of a non-linear system. Such an approach is classical and well known in the literature. Therefore, it would be worth considering whether a formal inclusion of non-linearities would not allow original control results to be obtained.

Control cycle repeatability tests have still not been carried out. What is the sampling time spread for different algorithm calculation frequencies? Is it known that the device actually works in the real time mode?

The text should be checked in detail. Incorrect reference to Algorithm 1 can still be found - there are "??" characters instead.

Reviewer 2 Report

attached
